# The Effect of a Handball Warm-Up Program on Dynamic Balance among Elite Adolescent Handball Players

**DOI:** 10.3390/sports10020018

**Published:** 2022-01-31

**Authors:** Abdolhamid Daneshjoo, Ali Hoseinpour, Hassan Sadeghi, Aref Kalantari, David George Behm

**Affiliations:** 1Department of Sport Injuries and Corrective Exercises, Faculty of Sport Sciences, Shahid Bahonar University of Kerman, Kerman 76169-13439, Iran; daneshjoo.hamid@uk.ac.ir (A.D.); hosseinpour70@yahoo.com (A.H.); aref07kalantari@gmail.com (A.K.); 2School of Human Kinetics and Recreation, Memorial University of Newfoundland, St. John’s, NL A1C 5S7, Canada; 3Department of Biomechanics and Sport Injuries, Faculty of Physical Education and Sport Sciences, Kharazmi University, Tehran 15719-14911, Iran; hasan.sadeghi81@gmail.com

**Keywords:** equilibrium, postural balance, risk factors, stability, instability

## Abstract

This study examined the effect of the handball warm-up program (HWP) on dynamic balance among elite adolescent handball players. In this case, 24 handball players were randomly assigned into experimental warm-up and control groups. The HWP was performed over 8 weeks (3 times per week). Dynamic balance before and after the intervention training programs were measured by the Biodex Balance System (BBS) and Y Balance test for the dominant (DL) and non-dominant legs (non-DL). After HWP training, BBS scores significantly improved in the overall (OSI) (30.4% and 31.1%), anterior-posterior (APSI) (44.6% and 35.2%), and medial-lateral stability indexes (MLSI) (38.8% and 43%) for both DL and non-DL. Post-training, the Y Balance test exhibited significant improvements in OSI (13.2% and 10.6%), anterior (17.2% and 12.6%), posteriolateral (12.8% and 11.3%), and posteriomedial stability indexes (9.2% and 7.9%) with DL and non-DL, respectively. In conclusion, dynamic balance improved overall after eight weeks of the HWP, with this improvement significantly greater with the trained versus the control group. Optimal balance during handball is an important factor to ensure coordinated and efficient movements and it is speculated that improved balance could positively impact injury prevention. Hence, the HWP program can be used as an alternative to a conventional warm-up program.

## 1. Introduction

Handball commenced in 1946 and is presently composed of almost 170 members of the International Handball Federation including 795,000 handball teams and an estimated 25 million players worldwide [1]. It is a dynamic and physically demanding sport, mainly because of the intensive body contact between the players. The necessity for high levels of agility, strength, and balance imposes a high demand on the musculoskeletal system. Handball is reported to sustain the highest relative incidence of sports injuries [2]. Injuries in team handball are common and result in high costs for the players, club, and public health system and may even cause long-term disability for the injured player [3]. The injury rate in youth handball players has been shown to be between 9.9 and 41.0 injuries per 1000 match h and between 0.9 and 2.6 injuries per 1000 training h [4]. Non-contact injuries are highly prevalent (68.5%) among male junior handball players (16 years old) [5]. In order to protect the players, an injury prevention program is an urgent need [2]. Unfortunately, there are only a few recent studies on the injury prevention with male handball players. It is reported that a multifaceted intervention program with an appropriate warm-up structure has been effective at enhancing the performance and preventing common injuries among players, especially soccer players [6,7,8].

Oliano and their colleagues [9] measured the effect of FIFA 11+ warm-up training program in addition to conventional handball training on postural balance of the knee joint in female handball athletes (aged 11–14 years). The intervention group performed the FIFA 11+ warm-up program, twice a week, with sessions lasting 40 min on average, for 12 weeks. The results demonstrated an improvement in postural balance (assessed through computerized dynamic posturography) in the intervention group, which did not occur in control group. Both groups showed improvements in knee isokinetic muscular power. In conclusion, the practice of FIFA 11+ in addition to conventional handball training demonstrated improvements in the postural balance of young players [9].

The role of balance is believed to be a critical component of neuromuscular control and as a modifiable risk factor contributes to limit mediolateral knee displacement and loading during dynamic activities. Balance is fundamental for people to cope with activities of daily living and to successfully engage in sport training such as; running and jumping [10]. Balance information is useful to identify players with an increased risk of sport injury and to decide if an injured player can return to sport without risk for re-injury [11,12]. Poor dynamic balance measured by a Y-balance test has been associated with greater risk of ankle sprain injury with male collegiate athletes [12]. Balance exercises seem to represent a key component of effective anterior cruciate ligament (ACL) injury prevention programs, which tend to focus on frontal plane knee control during static and dynamic tasks [13]. Intervention programs including balance, jumping, and agility have some positive effect in improving postural sway and dynamic balance in athlete [14,15].

The Handball warm-up program (HWP) is a multifaceted program and is a modified version of the FIFA 11+ warm-up injury prevention program based on the specific skills and demands of handball. HWP replaces the regular warm-up before technical and tactical drills. HWP consists of three parts included three levels of increasing difficulty including strength, balance, agility and jumping exercises. HWP also consists of a novel set of advanced running exercises that make it better suited as a comprehensive warm-up program for handball training and matches. In addition, the HWP is a feasible and easy program that can be incorporated into a small space. To date, little research exists quantifying balance measures as potential risk factors for handball players after multifaceted warm-up program. It is unclear whether a structured warm-up intervention program can enhance the dynamic balance among elite handball players. Hence, the aim of this study was to investigate the effect of the HWP on dynamic balance among elite, adolescent handball players. The first hypothesis was that implementing the HWP would improve dynamic balance scores in both DL and NDL of elite handball players. The second hypothesis was that implementing comprehensive HWP can enhance dynamic balance in both sagittal (anterior-posterior) and frontal (medial-lateral direction) planes.

## 2. Methods

### 2.1. Participants and Study Design

This experimental intervention study involved participants recruited among handball players who participated in the Iranian national league (with at least 5 years experiences). The 24 male handball players were randomly and equally divided into experimental (HWP: n = 12) and control (n = 12) groups (Table 1). Written informed consent from parents and coaches, as caretakers, on behalf of the minors was obtained. The inclusion criteria consisted of: handball players with an age of between 14–15 years old, 5 years of playing experience, no experience with the FIFA 11+ warm-up or similar exercise and had no previous experience using the Biodex Balance System, free of pain and lower limb injury or neurological impairment at the time of study. Participations were excluded from the study if they were involved in sports such as martial arts or dancing, which may affect balance abilities. Before participation, all procedures were explained to the participants, and they received an illustration of the HWP. Prior to starting the test session, all the players participated in familiarization sessions with balance tests. The procedure of this study was approved by the Institute of Research Management and Monitoring, University of Shahid Bahonar University of Kerman, Iran.

### 2.2. HWP Warm-Up Program

The HWP is a modified version of the FIFA 11+ warm-up injury prevention program specific to handball. HWP replaces the regular warm-up before technical and tactical drills. HWP consisted of three parts. The initial component is running exercises combined with dynamic stretching, lunges, walking, crawling side-to-side run, and smooth swaying carioca exercises with a partner. This component is performed in the width of a handball pitch with 5 cones, 4 m apart over 6 min (part I). The second part consisted of 6 exercises having three levels of increasing difficulty included strength, balance, agility, and jumping exercises (part II). The final component included advanced running exercises such as cutting and bounding with sudden changes in direction (part III) (Table 2). We encouraged the handball players to concentrate on the quality of their exercises. The different levels of difficulty improved the program’s efficiency and enabled coaches and players to individually adapt to the program. HWP duration was approximately 20–25 min, performed three times per week for eight weeks.

### 2.3. Control Group

We asked the control group to continue with their regular warm-up training during the season, which typically included running across the pitch and static stretching. In addition, the control group was promised that if the HWP was effective, they would receive the same program in eight weeks.

### 2.4. Measures

#### 2.4.1. Dynamic Single-Leg Balance Test

The Biodex Balance SD System (BBS) (Biodex Medical System Inc., Shirley, NY, USA) was used to measure dynamic single-leg balance. The three measured postural stability indices were overall stability index (OSI), medial-lateral stability index (MLSI) and anterior-posterior stability index (APSI). These indices indicated the standard deviation that assesses the sway displacement from the origin point of the platform. The foot displacement for the medial-lateral axes along the horizontal line as an x-direction, and the anterior-posterior axes along the vertical line as y-direction were evaluated as MLSI and APSI, respectively. With the integration of tilt degree in the anterior-posterior and medial-lateral direction, the OSI was calculated. All stability indices were recorded in degrees. Cachupe et al. (2001) showed the reliability measure across 3 test evaluations as 0.83 (OSI), 0.81 (MLSI) and 0.73 (APSI) [16].

Participants were asked to stand stationary on the BBS platform with their arms crossed over their chest. They were required to maintain their visual focus on the liquid crystal display. Participants were instructed to stand with the dominant leg (the leg preferred for ball kicking) as the supporting leg and the contralateral, non-dominant, leg was flexed at 90° [17]. The foot displacement was adjusted until a comfortable standing posture was achieved and simultaneously eased in controlling the cursor. The platform was then locked, and foot placement remained static throughout the test. The dynamic balance measures were recorded on Biodex level four. Participants were instructed to maintain the cursor at the origin point. Three trials were performed for 20 s each and 1 min of rest period between the trials. Three-minute rest was obtained between tests for each leg. The data was averaged to obtain the mean score for three trials. The non-supported leg was prohibited from resting on the supporting limb during the test. During the dynamic balance test, the laboratory was quiet.

#### 2.4.2. Y Dynamic Balance Test

The highly reliable (ICC ≤ 0.96) [18] Y Balance Test (measure of dynamic postural control) is performed on a grid of three lines. The dominant foot [19] was placed in the center of the grid, to equally bisect the antero-posterior and medial-lateral planes. Subjects were asked to reach as far as possible in the anterior, posterior medial (PMSI), and posterolateral (PLSI) directions, touching the line with the foot, and returning to a double-leg stance. The reaches were performed in a clockwise or counterclockwise direction, depending on whether the dominant leg was the right or the left, respectively. Hands were kept akimbo and the stance leg heel had to remain on the floor at all times. A warm-up of six practice trials was incorporated to overcome learning effects. Following a 5-min rest period, the test trials were conducted. Rest between trials was self-determined to that fatigue was minimized. Reach distances were recorded at the point of maximal reach from the center of the grid. The mean of the three reaches was normalized by dividing by the previously measured leg length to standardize the maximum reach distance ((excursion distance/leg length) × 100 = % maximum reach distance). An overall stability index (OSI) was calculated with the summation of the three directions. Greater excursion distances represented greater dynamic balance. Leg length was defined as the length measured from an anterior superior iliac spine to the medial malleolus of the tibia. Trials were deemed ineligible and repeated if the participant used the reaching leg to attain support, removed the supporting foot from the center of the grid, or was unable to maintain balance on the support leg throughout the trial.

### 2.5. Statistical Analysis

A 3-way (2 groups vs. 2 times vs. 2 legs) mixed model repeated measures ANOVA (IBM SPSS software; IBM Corp, Armonk, NY, USA) was employed to compare the variables between groups (HWP, control), times (pre-test, post-test), and legs. Data distribution was normally distributed (*p* > 0.05) and variance among groups was homogenous as assessed with the Shapiro-Wilk test and Levene’s test, respectively. The Bonferroni post-hoc test was conducted to detect the significant differences between groups for each test. The effect sizes were tested using partial eta (pη^2^) squared (0.01 = small effect, 0.06 = medium effect, and 0.14 = large effect) [20]. A significance level was considered at the 95% confidence level for all statistical parameters.

## 3. Results

Participant characteristics and demographic factors of participants were presented in Table 1. In addition, the t-test results show that there no significant difference between groups in the participants characteristics.

### 3.1. BBS Dynamic Single Leg Test

The results showed significant interactions between group and time with OSI (F_1,22_ =20.88, *p* < 0.0001, pη^2^ = 0.49), APSI (F_1,22_ = 59.71, *p* < 0.0001, pη^2^ = 0.73) and MLSI (F_1,22_ = 57.87, *p* < 0.0001, pη^2^ = 0.72). The Bonferroni post-hoc test showed significant pre- to post-test improvements with the HWP group for both the DL and non-DL in OSI (30.4% and 31.1%) APSI (44.6% and 35.2%), and MLSI (38.8% and 43%) (Table 3). However, the results did not show any differences with the control group in OSI, APSI, and MLSI (Table 2). There was a significant main effect for group (data combined over pre- and post-tests) with HWP exceeding control in OSI (F_1,22_ = 4.45, *p* = 0.046, pη^2^ = 0.17), APSI (F_1,22_ = 5.40, *p* = 0.030, pη^2^ = 0.19) and MLSI (F_1,22_ = 4.84, *p* = 0.039, pη^2^ = 0.18). A significant main time effect (data combined for HWP and control) demonstrated increases from pre- to post-test for OSI (F_1,22_ = 16.52, *p* = 0.001, pη^2^ = 0.43), APSI (F_1,22_ = 79.49, *p* < 0.0001, pη^2^ = 0.78) and MLSI (F_1,22_ = 32.16, *p* < 0.0001, pη^2^ = 0.59).

### 3.2. Y Dynamic Balance Test

The results showed significant interactions between group and time with the Y-test in OSI (F_1,22_ = 208.86, *p* < 0.0001, pη^2^ = 0.90), anterior (F_1,22_ = 77.17, *p* < 0.0001, pη^2^ = 0.78), PLSI (F_1,22_ = 68.15, *p* < 0.0001, pη^2^ = 0.76), and PMSI (F_1,22_ = 63.01, *p* < 0.0001, pη^2^ = 0.74). The Bonferroni post-hoc test showed significant pre- to post-test increases in the HWP group in OSI, anterior, PLSI, and PMSI (Table 4). However, the results did not show any significant differences with the control group with OSI, anterior, PLSI, and PMSI (Table 4). There was significant main effect for group differences with HWP exhibiting higher balance scores than control in OSI (F_1,22_ = 4.82, *p* = 0.039, pη^2^ = 0.18), anterior (F_1,22_ = 4.86, *p* = 0.038, pη^2^ = 0.18), PLSI (F_1,22_ = 4.37, *p* = 0.048, pη^2^ = 0.17), but no significant difference in PMSI (F_1,22_ = 2.31, *p* = 0.14). The results did reveal significant improvements (main effect for time, with data collapsed over HWP and control groups) between pre- and post-tests with the Y balance test for OSI (F_1,22_ = 264.32, *p* < 0.0001, pη^2^ = 0.92) by 13.2% and 10.6%, anterior (F_1,22_ = 104.07, *p* > 0.0001, pη^2^ = 0.83) by 17.2% and 12.6%, PLSI (F_1,22_ = 74.23, *p* < 0.0001, pη^2^ = 0.77) by 12.8% and 11.3%, and PMSI (F_1,22_ = 84.81, *p* < 0.0001, pη^2^ = 0.79) by 9.2% and 7.9% in DL and non-DL, respectively.

### 3.3. Comparison between Legs

For the BBS dynamic single leg balance test, there were significant main effect between legs in OSI (F_1,22_ = 9.44, *p* = 0.006, pη^2^ = 0.30), APSI (F_1,22_ = 65.07, *p*< 0.0001, pη^2^ = 0.75), but no significant differences in MLSI (F_1,22_ = 1.93, *p* = 0.178). Post-hoc tests showed significant increases with the DL in HWP group for APSI (*p* = 0.007) and with the control group with the Non-DL for OSI (*p* < 0.0001) as well as with the control DL for APSI (*p* < 0.0001). No significant difference in HWP group in OSI (*p* = 0.950). For Y balance test, results showed no significant main effect between legs in OSI (F_1,22_ = 4.05, *p* = 0.057), anterior (F_1,22_ = 3.758, *p* = 0.065), PLSI (F_1,22_ = 1.34, *p* = 0.259) and PMSI (F_1,22_ = 0.53, *p* = 0.476).

## 4. Discussion

The aim of this study was to investigate the effect of the HWP on dynamic balance among male adolescent, national calibre, handball players. The main results of this study showed that after an 8-week HWP, significant, large magnitude, effect size improvements were observed in dynamic balance with both BBS and Y balance tests. Therefore, the result of this study supports the hypothesis that a multifaceted, sport specific, warm-up program (HWP) can enhance balance in adolescent handball players. Balance during sport activities is an important factor for sport injury prevention. A decline in balance is seen following ankle and knee injuries [6], so it is suggested to use balance training to improve neuromuscular function and postural stability. Proper functional balance and postural control of the lower body are essential for both tactical and technical skill performance among handball players, and such attributes are assumed to reduce injury risk [21]. Warm-up before training may help to reduce risk of injury with sport [22].

HWP is a multifaceted exercise program, which includes dynamic stretching, strength, postural stability, plyometric, and balance components. Warm-up programs fundamentally include mild or moderate exercise types that are geared toward enhancing skill performance [23,24,25] by increasing blood flow, tissue elasticity, body temperature and axonal conduction velocity [26,27,28,29]. Warm-up activities may stimulate and increase muscle spindle sensitivity contributing to an improvement of postural stability [30]. Pasanen et al. (2009) and Daneshjoo et al. (2012) revealed that a warm-up program can improve neuromuscular function leading to better balance [6,31]. Leavey et al. (2010) reported that a 6-week multifaceted exercise program, which included strength and postural control balance components, improved dynamic balance in healthy men and women [32]. The 11+ warm-up injury prevention program, which is a multifaceted program, can improve dynamic balance in professional young male soccer players [6]. HWP is also a multifaceted pre-exercise program and this could contribute to improved dynamic balance in handball players. It was reported that exercises such as plyometric and single-leg standing enhance central nervous system (CNS) plasticity and improve the function of the balance feedback loop over other balance systems [33]. Down-regulation of the H-reflex (spinal reflexes associated with afferent excitability of the spinal motoneurons) after a training program that included balance exercises can improve dynamic balance by preventing reflex-mediated joint oscillations [14]. Such a training program may alter feedback of mechanoreceptors and improve sensorimotor integration consequently leading to alterations of the motor response (adaptations of neuromuscular control) [14].

Olmsted et al. (2003) found that the supporting leg requires ankle dorsiflexion, knee, and hip flexion with an adequate range of motion, strength, proprioception, and neuromuscular control to perform the Y dynamic balance test [34]. Earl and Hertel (2001) affirm that adequate strength of lower extremities is necessary to achieve dynamic balance [35]. The findings from this study make several contributions to the current literature. The HWP is a feasible, easy to perform program that can be incorporated into a small space. Moreover, the intensity of HWP program can be manipulated to ensure progression or an appropriate preparation dependent on the level of the athlete. One of the reasons for the effectiveness of the HWP exercise program is the type of exercises used in this program. Factors that can contribute to the effectiveness of an exercise program to enhance performance include active stretching exercises, strengthening core or trunk stability, strength, balance, and players’ awareness of proper body alignment in various movements (proprioception), especially when jumping and landing. The dynamic balance enhancement seen in the HWP group can be attributed to the thrice weekly, 8-week, exercise programs’ dosage [33].

The result showed dynamic balance improvements in both anterior-posterior and medial- lateral directions. The HWP is multifactorial program, which included sagittal plane exercises as well as exercises such as single leg jump and cutting exercises, which may have contributed to balance improvements in the medial-lateral direction. Based on the concept of training specificity [36], training movements and velocities must closely mimic the sport action or task. Hence, to improve performance in a complex multi-directional sport such as handball, plyometric training must incorporate explosive (high velocity) anterior-posterior and medial-lateral movements. Sport specific programs such as HWP should enhance lower extremity balance profiles, contributing to an improved base of support (postural stability) from which to transition rapidly in multiple directions [37].

The results showed no significant pre-test to post-test differences between legs with the dynamic balance tests after an 8-week HWP. The results mainly suggests that a functional symmetry exists between the dominant and non-dominant legs in dynamic balance tests among adolescent handball players. These findings are in agreement with some of the research that examined single-leg stance performance in healthy subjects [38,39]. The present study is in accord with these previous results and further indicates that balance performance symmetry also exists for dynamically challenging tasks in adolescent handball players. Balance performance symmetries between the dominant and non-dominant legs can be attributed to the physical activity patterns of the subjects in this study. The dominant leg is used to initiate the jump, while the non-dominant leg primarily provides postural support during landing [40], while actively generating the knee and hip flexion and extension that are required to support the individual’s weight [41].

Finally, there are a few limitations need to be considered. First, the sample size was small and future studies need to consider a larger sample size. Second, the researchers could not conduct follow-up tests over an extended duration (e.g., two or three months). More research is required to determine the persistence of the intervention protocols over an extended period.

## 5. Conclusions

It can be concluded that HWP improved dynamic balance among elite, male adolescent, handball players with no significant difference between legs. The results of the present study support that handball training programs and competitions (control group) did not negatively affect the dynamic balance of adolescent players. It is suggested that the HWP could be implemented and incorporated into regular handball practice before starting technical and tactical drills. Future studies are also needed to investigate the impact of HWP on other physical performance skills and modifiable risk factors of handball players. The findings of this research can be helpful for coaches and trainers who can strategize training programs to improve balance of elite, adolescent handball players. Although not monitored in this research, higher levels of dynamic balance are associated with lower incidence of injuries [11,12,13,21] and thus balance training should be incorporated to not only enhance performance [10,14] but also reduce injury incidence.

A handball-specific multifaceted warm-up program (HWP) which combines strength, neuromuscular control, balance and activity specific motion patterns without using special equipment can be incorporated instead of conventional warm-up program to improve dynamic balance among adolescent handball players. Playing handball did not negatively affect the dynamic balance of adolescent players.

## Figures and Tables

**Table 1 sports-10-00018-t001:** Participant characteristics (mean ± SD).

	HWP Group	Control Group	*p*-Value
Age (y)	14.75 ± 1.14	14.58 ± 0.51	0.65
Weight (kg)	65.87 ± 9.60	68.42 ± 12.61	0.2
Height (m)	1.77 ± 0.50	1.78 ± 0.69	0.68
Leg length (cm)	94.73 ± 3.29	95.50 ± 5.20	0.67
BMI (kg/m^2^)	20.85 ± 2.64	21.46 ± 3.62	0.24

Note. y = year; kg = kilogram; cm = centimeter; kg/m^2^ = kilogram per square meter; *p*-value = independent *t*-test value among groups.

**Table 2 sports-10-00018-t002:** Handball warm-up program (HWP), exercises in the structured warm-up program.

Exercise	Repetitions	Total Time
I. Running exercises and brisk walking, 8 min (opening warm up, in pairs; course consists of 6–10 pairs of parallel cones):		
Running, forward and backward	2	60 s
Lunge with turning upper body	2	90 s
Walking with reach hand to opposite leg	2	80 s
Crawling	2	90 s
Side-to-side run	2	100 s
Smooth swaying carioca	2	50 s
II. Strength, plyometric, balance, 10 min (one of three exercise progression levels each training session):		
Crossover V sit-ups:		
Level 1: knee and elbow in flexion	2 × 10	50 s
Level 2: knee and elbow full extension	2 × 10	50 s
Level 3: knee and elbow full extension and elevate upper body (trunk flexion)	2 × 10	60 s
Back extension		
Level 1: Back extension	2 × 10–15 s (each side)	80–120 s
Level 2: Back extension	2 × 15–20 s (each side)	80–120 s
Level 3: Back extension	2 × 20–25 s (each side)	80–120 s
hamstrings strength		
Level 1 Hamstrings curl	3–5	15–20 s
Level 2 Nordic hamstrings	4–6	25–30 s
Level 3 Nordic hamstrings	7–10	40–45 s
Single leg balance		
Level 1: single leg stance with pass ball	2 × 20 s (each leg)	80 s
Level 2: single leg with heel raises with pass ball	2 × 20 s (each leg)	80 s
Level 3: single leg with jumping with pass ball	2 × 20 s (each leg)	80 s
Medicine ball throwing		
Level 1: throwing with two hands	4–6	30 s
Level 2: throwing with horizontal jumping	4–6	30 s
Level 3: throwing with horizontal jumping	4–6	30 s
Ladder training		
Level 1: double leg jump	2	30 s
Level 2: single leg jump	2	35 s
Level 3: fast single leg jump	2	35 s
III. Cutting and Bounding with stop exercises, 4 min		
Cutting training		
direct 8 m, oblique 2 m	3	30 s
direct 12 m, oblique 4 m	3	35 s
direct 18 m, oblique 6 m	3	40 s
Bounding	2	30 s

Note. s = second; m = meter.

**Table 3 sports-10-00018-t003:** Athlete single leg mean (SD); BBS dynamic balance values.

	Group	Leg	Pre-Test	Post-Test	∆%	(95% CI)	*p*-Values
OSI	HWP	DL	2.76 (1.11)	1.92 (0.44)	30.4	(0.21 to 1.48)	*p* < 0.013
		Non-DL	2.76 (0.96)	1.90 (0.83)	31.1	(0.61 to 1.11)	*p* < 0.0001
	Control	DL	3.33 (0.89)	3.42 (1.11)	−2.6	(−0.42 to 0.23)	NS
		Non-DL	2.64 (0.79)	2.65 (0.82)	−0.4	(−0.24 to 0.22)	NS
APSI	HWP	DL	2.42 (0.49)	1.34 (0.35)	44.6	(0.78 to 1.38)	*p* < 0.0001
		Non-DL	1.90 (0.40)	1.23 (0.26)	35.2	(0.47 to 0.86)	*p* < 0.0001
	Control	DL	2.55 (0.64)	2.36 (0.63)	7.4	(−0.01 to 0.39)	NS
		Non-DL	1.82 (0.55)	1.88 (0.55)	−3.2	(−0.20 to 0.07)	NS
MLSI	HWP	DL	1.98 (1.09)	1.21 (0.75)	38.8	(0.45 to 1.08)	*p* < 0.0001
		Non-DL	1.93 (0.64)	1.10 (0.40)	43.0	(0.54 to 1.12)	*p* < 0.0001
	Control	DL	2.31 (0.95)	2.32 (0.84)	−0.4	(−0.15 to 0.12)	NS
		Non-DL	1.88 (0.64)	2.10 (0.67)	−11.7	(−0.46 to 0.24)	NS

Note. OSI = Overall stability index; APSI = anterior posterior stability index; MLSI = medial lateral stability index; DL = dominant leg; Non-DL = non-dominant leg; NS: non-significant; *p*-values = differences between pre-test and post-test.

**Table 4 sports-10-00018-t004:** Y dynamic balance test; values are mean (SD) and percentage of change (∆) [values are mean (95% CI)] from post to pre time points of dominant to non-dominant legs.

	Group	Leg	Pre-Test (cm)	Post-Test (cm)	∆%	(95% CI)	*p*-Values
OSI	HWP	DL	88.7 (4.4)	100.4 (2.7)	13.2	(9.6–13.7)	*p* < 0.0001
		Non-DL	89.6 (4.4)	99.1 (4.0)	10.6	(8.2–10.8)	*p* < 0.0001
	Control	DL	91.7 (4.9)	92.5 (4.4)	0.9	(−0.5–2.1)	NS
		Non-DL	89.4 (5.2)	89.8 (5.1)	0.4	(−0.2–1.1)	NS
Anterior	HWP	DL	84.3 (5.6)	98.8 (3.0)	17.2	(10.7–18.4)	*p* < 0.0001
		Non-DL	85.4 (4.9)	96.2 (3.7)	12.6	(7.7–13.9)	*p* < 0.0001
	Control	DL	88.8 (4.9)	91.1 (5.8)	2.6	(−0.3–4.8)	NS
		Non-DL	86.9 (4.6)	86.6 (3.9)	-0.3	(−2.3–1.5)	NS
Posterior-L	HWP	DL	88.7 (4.9)	100.4 (4.3)	12.8	(8.7–14.7)	*p* < 0.0001
		Non-DL	90.1 (7.0)	100.3 (5.6)	11.3	(7.4–13.1)	*p* < 0.0001
	Control	DL	92.5 (6.5)	91.8 (6.5)	−0.7	(−3.1–1.7)	NS
		Non-DL	88.7 (6.6)	88.5 (5.6)	0.2	(−0.02–2.3)	NS
Posterior-M	HWP	DL	93.2 (4.8)	101.8 (4.7)	9.2	(5.7–11.6)	*p* < 0.0001
		Non-DL	93.2 (5.7)	100.6 (6.5)	7.9	(4.8–10.0)	*p* < 0.0001
	Control	DL	93.6 (7.1)	94.5 (7.3)	0.9	(−0.5–2.2)	NS
		Non-DL	92.7 (8.2)	93.1 (9.4)	0.4	(−1.2–1.9)	NS

Note. DL = dominant leg; Non-DL = non-dominant leg; cm = centimeter; Posterior-L = posterior lateral; Posterior-M= posterior medial; NS: non-significant.

## Data Availability

All data generated or analyzed during this study are included in this published article.

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
