# Peer review of "The Effect of a Handball Warm-Up Program on Dynamic Balance among Elite Adolescent Handball Players"

_sports, 2022, doi:10.3390/sports10020018_

Round 1

Reviewer 1 Report

Thank you for the opportunity to review this publication with the aim to examine effects of a handball warm-up program (HWP) on dynamic balance among elite adolescent handball players. The question of the study is definitely interesting for a broad target group of athletes, caregivers, sports scientists and medical professionals.

The results of the studies largely coincide with the data available so far and could shed more light on the area.

The article provides an interesting insight on the topic. However, concerns on some key points are listed below:

  1. The introduction is well structured but must be scientifically reinforced. The authors base their research on the incidence of injury in the manuscript by Olsen et al. and in the explanation of the work by Oliano et al. However, the fact that the two concepts may be related does not mean that there are other factors that can lead to injuries. It is necessary to clarify well if the justification of the article is to improve the harmful incidence or if it is to seek improvement in the coordination and efficiency of the movement. Also reinforce with more relevant studies the relationship between the concepts of dynamic balance and injuries.
  2. The authors have made it clear that they are doing this study because there is little scientific evidence on the subject, but this is a general comment. It is necessary to emphasize more clearly and practically what this study contributes to the scientific and practical community.
  3. Provide hypothesis/hypotheses.
  4. The procedure of this study was approved by the Institute of Research Management and Monitoring, University of Shahid 94 Bahonar University of Kerman, Iran. But, the registration number of the ethics committee is not indicated.
  5. There is no statement of parental consent concerning the underage players participated in the study.
  6. SPSS manufacturer details: IBM Corp., Armonk, NY.
  7. Effect size is not referenced
  8. A linguistic revision, and review of formal aspects, would be desirable.
  9. They do a complete and orderly discussion but little and repetitive in terms of references. However, the most important thing is that they emphasize and explain the reasons for the differences found. Currently there are many articles on balance that can help you complete both introduction and discussion, and they do not have to be solely about handball as the content can be extrapolated. See how these sample articles complete the information from their study and comparing it with other sports and the importance of balance in sport:
  • Comparative study of stabilometric parameters in sportsmen of various disciplines.
  • Balance training for neuromuscular control and performance enhancement: A systematic review
  • Stabilometry profile in fixed seat rowers.
  • Relationship between balance ability, training and sports injury risk.
  • The Effect of Balance and Sand Training on Postural Control in Elite Beach Volleyball Players
  • Effects of Balance Training on Balance Performance in Youth: Are There Age Differences?

Author Response

See attached response.

Reviewer 2 Report

Table 1 provided information about handball warm-up program, not about the participants, so why authors linked to this table in line 84?

Authors should provide information regarding the gender of the participants in section "Participants and study design".

Item 21 in bibliography should be corrected - the lack of authors' last name.

Author Response

See attached response.

Round 2

Reviewer 1 Report

The authors have further revised the resubmitted manuscript. The recommendations and content-related criticisms of the reviewers have largely been incorporated. 

 Some minor issues are pointed below.

L71 - Make a more detailed hypothesis, it is obvious that if a previous HWP is made, the dynamic balance scores will be increased

L78 - Language review is required.

L104 - All tables must be in the same format, this table does not meet the standards of the journal.

L227 - The discussion has been strengthened, but it is still one of the weak points of the manuscript. It would be appreciated if the comparison of results as well as the interpretation by the authors were increased.

L307 - Please check that all references meet the journal criteria and are in the same format.

I would like to express my gratitude for the opportunity to review this manuscript and recommend acceptance of the manuscript after linguistic revision.

Author Response

See attached file.

Please thank the reviewer for their contributions to improving the manuscript.
